# Prognostic Value of the Time-to-Positivity in Blood Cultures from Septic Shock Patients with Bacteremia Receiving Protocol-Driven Resuscitation Bundle Therapy: A Retrospective Cohort Study

**DOI:** 10.3390/antibiotics10060683

**Published:** 2021-06-08

**Authors:** Hong-Jun Bae, June-sung Kim, Muyeol Kim, Youn-Jung Kim, Won Young Kim

**Affiliations:** Asan Medical Center, Department of Emergency Medicine, College of Medicine, University of Ulsan, Seoul 05505, Korea; mtation@naver.com (H.-J.B.); jstyle06@amc.seoul.kr (J.-s.K.); kurt0217@naver.com (M.K.); yjkim@amc.seoul.kr (Y.-J.K.)

**Keywords:** sepsis, septic shock, time-to-positivity, bacteremia, mortality

## Abstract

Introduction: To evaluate the prognostic value of the time-to-positivity in patients with culture-positive septic shock. Methods: Retrospective study using a prospective data registry was performed at the emergency department of a tertiary hospital. Consecutive adult patients with septic shock (N = 2499) were enrolled between 2014 and 2018. Bacteremia was defined using blood cultures, and viral and fungal pathogens were excluded. The primary outcome was the 28-day mortality. Results: In 803 (46.7%) septic shock patients with bacteremia, median TTP was 10.1 h. The most prevalent isolated bacterial pathogens were *Escherichia coli* (40.8%) and *Klebsiella* (23.4%). Although the TTP correlated with a higher sequential organ failure assessment score (Spearman’s rho = −0.12, *p* < 0.01), it showed no significant difference between the 28-day survivors and non-survivors (10.2 vs. 9.4 days, *p* = 0.35). In subgroup analysis of the *Escherichia coli* and *Klebsiella* bacteremia cases, a shorter TTP showed prognostic value for predicting the 28-day mortality. The optimal TTP cut-off for *Escherichia coli* and *Klebsiella* was 10 h and 8 h, respectively. Conclusions: The prognostic value of the TTP in septic shock patients receiving bundle therapy may be limited and its clinical interpretation should only be made on a pathogen-specific basis.

## 1. Introduction

Sepsis is a life-threatening condition involving organ dysfunction caused by a dysregulated host response to infection [1]. It is a common cause of ICU admission and has a high mortality rate [2]. Blood cultures, as the gold standard test, are necessary for the diagnosis of bacteremia in septic shock patients [3] and current guidelines recommend at least two sets of cultures being tested as a resuscitation bundle [4]. The time-to-positivity (TTP) of a blood culture can now be accurately determined due to improved microbiological testing methods. This parameter is defined as the period from the beginning of culture incubation to the detection of bacterial growth using an automated system; it has been suggested as a surrogate marker for bacterial concentration in the blood [5,6] and as a diagnostic and prognostic tool [7,8,9,10]. As quantitative blood cultures are not easy to conduct as a routine test, and infection markers such as C-reactive protein and procalcitonin have limited prognostic value, the TTP represents a potent new prognostic indicator of the bacterial load in patients with septic shock. Previous studies have demonstrated the value of the TTP as a prognostic tool under specific conditions such as neutropenia or for specific pathogens including *Staphylococcus aureus* [11], *Pseudomonas aeruginosa* [9], *Klebsiella* pneumonia [12], and even candidemia [13,14]. Notably however, the relationship between the TTP and clinical outcomes in septic shock patients receiving protocol-driven resuscitation bundle therapy at an emergency department (ED) has rarely been investigated. Further evidence is therefore needed to validate the utility of TTP as an accurate clinical indicator among septic shock patients.

We hypothesized that a shorter TTP, reflecting a higher bacterial load, would be associated with poorer clinical outcomes and could thus be used as a prognostic marker in septic shock. We tested this in our current study cohort of septic shock patients with bacteremia.

## 2. Material and Methods

### 2.1. Study Design and Setting

This was a single-center, observational, retrospective cohort study conducted in the ED of Asan Medical Center, Seoul, Republic of Korea, an urban tertiary referral hospital that treats more than 120,000 patients annually. Data were collected from the prospective septic shock registry of our hospital for adult patients treated between 1 January 2014 and 31 December 2018. This study was approved by the Research Ethics Committee of Asan Medical Center (No. 0548), which waived the requirement for patient informed consent.

### 2.2. Study Population and Variables

All adult septic shock patients (i.e., aged ≥ 18 years) diagnosed at our ED are enrolled in a prospective registry unless they refused the recommended treatment, had a “do not resuscitate” order, required a transfer to another hospital due to bed availability, or had been transferred from another hospital after appropriate resuscitation [15,16]. In accordance with the previously established definition of septic shock, the cases enrolled in our hospital’s registry were patients clinically assessed as having an infectious condition with refractory hypotension (mean arterial pressure ≤ 65 mmHg), requiring vasopressors despite adequate fluid infusion, or those with blood lactate levels of at least 4 mmol/L (Sepsis-2) [17]. Based on the Surviving Sepsis Campaign guidelines [4], all of our current study patients had been treated using protocol-driven resuscitation bundle therapy, which included aggressive fluid resuscitation, vasopressor use, blood cultures, empirical antibiotic regimens, and serial lactate measurements within 3 h of shock recognition.

Registry data were analyzed for the initially screened adult septic shock patients who received resuscitation bundle therapy and only the bacteremia cases were finally included in our present study cohort. Bacteremia was defined as a blood culture-confirmed and documented infection by a bacterial strain. Patients with microbiologically proven viral, fungal, or other parasitic infections were excluded from the analyses.

Blood cultures were routinely obtained from at least two or more different anatomical sites within 30 min after emergency department arrival. Each blood sample of 10 mL was immediately cultivated to both aerobic and anaerobic bottles (BACTEC, BD Diagnostics, Erembodegem, Belgium). One blood sample was always obtained via an indwelling catheter, if present. Broad-spectrum empirical antibiotics were administered as soon as possible after appropriate culture samples had been obtained. The time period between blood culture sample incubation and the detection of a positive signal was defined as the TTP. If a positive signal was detected in both aerobic and anaerobic bottles, the shorter TTP was used in the analysis. The sites of infection were clinically assessed by the primary physician in each case from the patient’s history, physical examination results, and laboratory and imaging data.

The collected data for the patients in our septic shock cohort included demographics, underlying medical illnesses, initial vital signs, and clinical outcomes such as intensive care unit (ICU) admission, requirements for mechanical ventilation and renal replacement therapy, and 28-day mortality. The date of death for each of the non-survivors was obtained from the South Korean National Health Insurance Service records. Laboratory data were also collected in our current analyses, including white blood cell counts, hemoglobin, prothrombin time (international normalization ratio), lactate, and C-reactive protein. Sequential organ failure assessment (SOFA) scores were calculated based on the initial clinical and laboratory data on admission.

The primary outcome in our current investigation was the 28-day mortality and secondary outcomes included the sequential organ failure assessment (SOFA) score and ICU admission. Additional subgroup analysis was performed in the septic shock patients harboring the most common bacterial pathogens i.e., *Escherichia coli* and *Klebsiella* species.

### 2.3. Statistical Analysis

Descriptive statistics were stratified by the 28-day mortality outcomes (i.e., survivors and non-survivors). The continuous variables in the baseline demographics, clinical characteristics, laboratory data, and outcomes were presented as medians with interquartile range (IQR) and as a frequency with a category percentage. The Kolmogorov-Smirnov test was used to validate the normality of distribution. Categorical variables were assessed using the Fisher’s exact or chi-squared tests. The Spearman rank correlation coefficient was used to analyze the association between the SOFA score and TTP. For subgroup analysis of patients infected by the *Escherichia coli* and *Klebsiella* bacterial strains, Kaplan-Meier survival curves with a log-rank test were used to compare these two groups. A univariate logistic regression test was employed to identify risk factors that are predictive of a 28-day mortality in bacteremia septic shock patients, including the variables that had shown differences between the survivor and non-survivor groups. To subsequently control for confounders, a Cox proportional hazard model was constructed for the variables that showed a significant association in the univariate logistic regression model.

To compare the prognostic utility of the TTP among specific bacterial pathogens, a receiver operating characteristic (ROC) curves were generated and the areas under the curve (AUCs) were calculated in each case. The optimal cutoff values for these metrics were determined using the Youden index (sensitivity + specificity − 1), followed by a computation of the sensitivity and specificity using a standard statistical formula. *p*-values lower than 0.05 were considered to indicate statistical significance in this study. All statistical analyses were performed using SPSS Statistics for Windows, version 26.0 (IBM Corp., Armonk, NY, USA).

## 3. Results

### 3.1. Study Population and Blood Culture Findings

During the assigned study period, 2499 adult septic shock patients were treated in the ED of our hospital. We excluded 781 of these patients from further analysis due to a transfer from/to other hospitals (N = 218), a ‘do not resuscitate’ (DNR) order (N = 298), a refusal of treatment (N = 89), and infection with viral or fungal pathogens (N = 176). The remaining 1718 cases were initially enrolled in our septic shock cohort. Among these cases, 803 (46%) patients with documented bacteremia were finally included in our present analyses (Figure 1). The median age of this population was 67.0 years (range, 59.0–75.0) and 474 patients (59.0%) were male. The overall 28-day mortality rate was 16.1%.

The baseline characteristics of our final study population according to the 28-day mortality outcomes (i.e., survivors and non-survivors) are presented in Table 1. There were no differences in gender or age between the survivor and non-survivor groups. No differences in comorbidities were found either other than coronary artery disease (8.8 vs. 15.5%, *p* = 0.02; survivor vs. non-survivor, respectively). Septic shock from the lower respiratory tract was associated with a poor 28-day mortality outcome (10.4 vs. 39.2%, *p* < 0.01). On the other hand, patients with a urinary tract (19.6 vs. 8.5%, *p* < 0.01) or hepato-biliary-pancreas focus (47.8 vs. 31.0%, *p* < 0.01) tended to have a better prognosis. The respiratory rate was similar between the groups (20.0 vs. 20.0%, *p* < 0.01), but the body temperature was higher in the survivor group (38.0 vs. 37.4 °C *p* < 0.01). In the laboratory results, the white blood cell counts (9.1 vs. 8.1 × 10^3^/μL, *p* = 0.02) and hemoglobin level (10.8 vs. 10.0 g/dL, *p* < 0.01) were higher in the survivor group. Notably however, prothrombin (1.3 vs. 1.4, *p* < 0.01), initial lactate (3.1 vs. 4.7 mmol/L, *p* < 0.01), blood urea nitrogen (24.0 vs. 31.0 mg/dL, *p* < 0.01), and creatinine (1.3 vs. 1.7 mg/dL, *p* < 0.01) were higher in the non-survivor group.

Although the non-survivor group showed a higher SOFA score (7.0 vs. 11.0, *p* < 0.01), there was no significant difference in the TTP (10.2 vs. 9.4 h, *p* = 0.35) between the groups. Source controls were not frequently performed in the non-survivor group (38.0 vs. 47.6%, *p* = 0.04). The ICU admission rate (59.7 vs. 47.5%, *p* = 0.01), use of a mechanical ventilator (55.0 vs. 14.8%, *p* < 0.01), and renal replacement therapy (41.9 vs. 10.2%, *p* < 0.01) were more frequent in the non-survivor group (Table 1).

Gram-negative bacteria were predominantly isolated from our current study patients (Table 2), with *Escherichia coli* found to be the most common cause of the bacteremia, followed by *Klebsiella* strains (including *Klebsiella pneumoniae*, *oxytoca*, and *ornithinolytica*).

### 3.2. Risk Factors fora 28-Day Mortality in Septic Shock Patients with Bacteremia

The results of our univariate and multivariate logistic regression analyses to explore and identify risk factors for a 28-day mortality in our bacteremia patients are presented in Table 3. Cox regression analysis demonstrated that bacteremia of a urinary tract origin (HR 0.42; CI 0.20–0.88, *p* = 0.02) or hepato-biliary-pancreatic origin (HR 0.48; CI 0.29–0.79, *p* < 0.01) was associated with a lower 28-day mortality. With respect to the laboratory results, higher lactate levels were associated with a higher mortality rate (HR 1.18; CI 1.09–1.26, *p* < 0.01). Moreover, lower levels of hemoglobin (HR 0.87; CI 0.79–0.96, *p* < 0.01) and a higher SOFA score (HR 1.17; CI 1.09–1.25, *p* < 0.01) were associated with poorer clinical outcomes in a multivariate logistic regression model.

### 3.3. Time-to-Positivity in the Septic Shock Patients with Bacteremia

The median TTP in our blood culture-positive septic shock patients was 10.1 h (IQR, 8.2–13.3 h). The TTP distribution for our study patients is shown in Figure 2. Even though the TTP was found not to differ between the survivor and non-survivor groups (10.2 vs. 9.4 h, *p* = 0.35), a shorter TTP did show a weak negative correlation with a higher SOFA score (Spearman’s rho = −0.10, *p* < 0.01; Figure 3).

### 3.4. Subgroup Analysis of the Septic Shock Patients with Escherichia coli or Klebsiella bacteremia

When stratified by the isolated bacterial pathogen, we found that the TTP in the *Escherichia*
*coli* or *Klebsiella* bacteremia subgroups showed a significant difference between the survivors and non-survivors. The baseline characteristics of the *Escherichia coli* and *Klebsiella* bacteremia patient populations are presented in Appendix A, respectively. The TTP was shorter in the non-survivors in both of these subgroups (9.8 vs. 8.6 h, *p* = 0.01 for *Escherichia coli*, Appendix A) (9.8 vs. 8.6 h, *p* = 0.03 for *Klebsiella* species, Appendix A). In septic shock cases with *Escherichia coli* bacteremia, ROC analysis revealed that a cut-off of 10 h had the best sensitivity and specificity for predicting the 28-day mortality (84.4% and 42.0%, respectively). The AUC was 0.64 (95% CI 0.55–0.73, *p* = 0.01; Figure 4) and a TTP shorter than 10 h was associated with a 28-day mortality outcome in this group (OR 4.06; 95% CI 1.51–10.88, *p* < 0.01). In the *Klebsiella* bacteremia subgroup, we found that the optimal cut-off for the TTP was 8 h, with a 34.5% sensitivity and 80.9% specificity. The AUC in this case was 0.62 (95% CI 0.48–0.77, *p* = 0.07; Figure 4). However, a TTP shorter than 8 h was not significantly associated with a 28-day mortality in this group (OR 2.22; CI 0.94–5.26, *p* = 0.07).

## 4. Discussion

Our current study findings indicate a median TPP in adult culture-positive septic shock patients of 10.1 h (IQR 8.2–13.3). We further found that even though a shorter detection time has limited prognostic value in predicting a short-term mortality in septic shock patients, the association between the TTP and SOFA score suggested that a shorter TPP may be associated with a higher clinical severity in septic shock patients.

Few studies to date have investigated the value of the TTP in overall bacteremia or sepsis populations, and the reported data have been limited and vary greatly. The average TTP in one such study of an adult blood stream infection population was much longer than our current result at 24.91 h (95% CI 2.2–47.62) [7]. However, another prior multi-center study of pediatric sepsis patients reported a comparable median TTP of 11.7 h (IQR 8.3–17.1). These discrepancies have a number of plausible explanations. First, our current study population included only septic shock cases and the higher bacterial load in these patients may have contributed to the shorter TTP. Moreover, the causative pathogens will differ between study populations. As one example of this, a previous study of the clinical utility of the TTP included fungal pathogens, which have a much longer detection time in blood cultures than gram-negative bacteria [18].

In support of our present findings, the TTP was found in a previous report not to predict poor outcomes in pediatric sepsis cases [19]. In addition, a prior retrospective study in an adult bloodstream infection population reported that the TTP in mixed cultures had no prognostic significance [7]. When patients stratified by the source of infection, such as lung infection, urinary tract infection and hepato-biliary-pancreas infection, the difference in TTP between survivors and non-survivors was not significant in each group. (12.1 vs. 11.8 h, *p* = 0.73 for lung infection; 10.8 vs. 9.4 h, *p* = 0.73 for urinary tract infection; 9.9 vs. 9.0 h, *p* = 0.35 for hepato-biliary-pancreas infection, Appendix A). This result may have been due to the heterogeneity of bacterial pathogens. The value of an early TTP as an indicator of poor outcomes for single specific bacterial pathogens, such as *Escherichia coli* [20], various *Klebsiella* species [12], and *Pseudomonas aeruginosa* [9] had already been documented in both adults and children. In line with such previous studies, we here found that an early TTP in specific pathogen subgroups, i.e., patients with septic shock caused by *Escherichia coli* and *Klebsiella* bacteremia, was significantly associated with the 28-day mortality. It should be noted in this regard, however, that the TTP varies by pathogen and also the focus of infection [18]. Sepsis is a heterogeneous syndrome caused by numerous different pathogens and this heterogeneity and differences in growth times will necessarily obfuscate the significance of the TTP in septic shock cases.

Of relevance to the interpretation of our current findings and previous evidence, although the TTP provides indirect information regarding a pathogenic biomass, reflecting both the bacteremia load and microbial growth rate, the relationship between the TTP and mortality outcomes is not always linear. This may have had an important bearing on the apparent prognostic value of the TTP. For example, although most prior studies have found that a shorter TTP is associated with adverse clinical outcomes, as evidenced for example in a report on *Staphylococcus aureus* bacteremia cases [21], a large retrospective study from Canada found that a delayed TTP may be associated with an increased mortality [22]. Another retrospective study of a bloodstream infection group, in which both gram-positive and -negative bacteria were involved, classified the study population into short (<12 h), medium (≥12 h and ≤27 h), and long (>27 h) TTP groups, and found that both a long and a short TTP was significantly associated with a higher mortality [10]. Non-significant differences were found between the clinical characteristics of the patients with the short and long TTP values in that study. These findings may suggest that the TTP is not a linear risk factor in terms of mortality, and this may underlie our present findings.

Our current study had some potential limitations of note. As a single-center study, our results may lack general applicability. Though no differences in comorbidities other than coronary artery disease were found between survivors and non-survivors, we found that patients with underlying malignant disease or liver cirrhosis have higher incidence of bacteremia (43.0 vs. 48.8%, *p* = 0.03 for malignancy; 12.7 vs. 18.7%, *p* < 0.01 for liver cirrhosis, no bacteremia vs. bacteremia, respectively, Appendix A). Generally, comorbidities such as malignancy and liver cirrhosis tend to show worse prognosis in infectious condition. Further study is needed in this subgroup. In addition, although known prognostic indicators for sepsis patients were included in our multivariate analysis, confounders may still exist. Both the TTP and clinical outcomes could be influenced by patient factors, such as immunosuppression and recent antibiotic use.

## 5. Conclusions

Although the TTP may correlate with clinical severity, it is not a good predictor of the short-term outcomes in patients with culture-positive septic shock. However, in cases of bacteremia caused by a specific pathogen, TTP may have prognostic value in predicting short-term mortality.

## Figures and Tables

**Figure 1 antibiotics-10-00683-f001:**
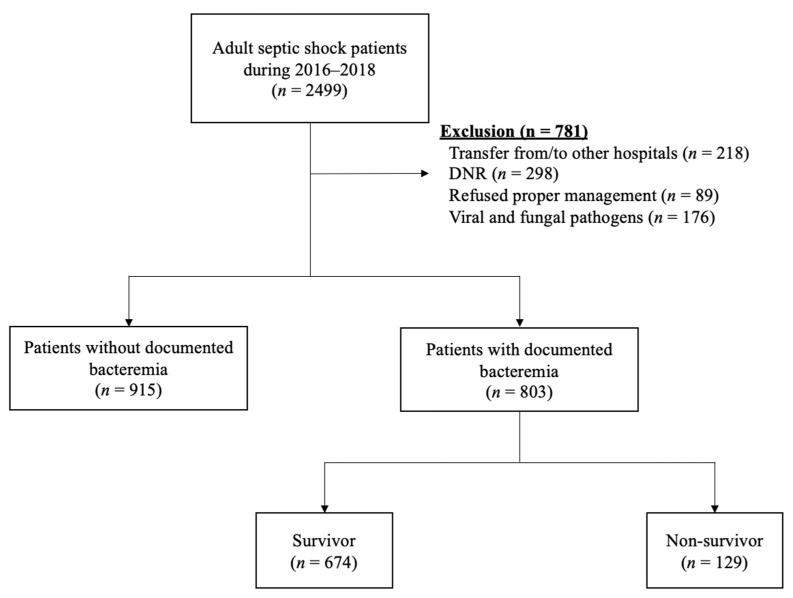
Flowchart of the study population. Abbreviations: DNR = Do not resuscitate.

**Figure 2 antibiotics-10-00683-f002:**
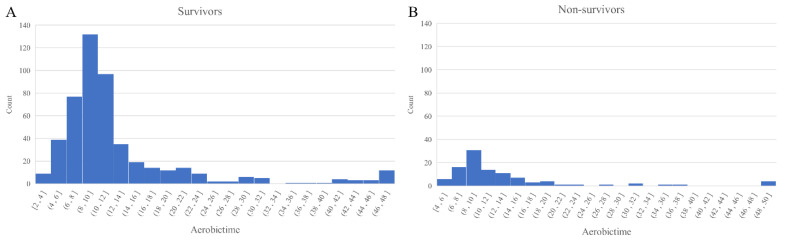
Distribution of the TTP values according to the 28-day mortality ((**A**) survivors and (**B**) non-survivors). Abbreviations: TTP = Time-to-positivity.

**Figure 3 antibiotics-10-00683-f003:**
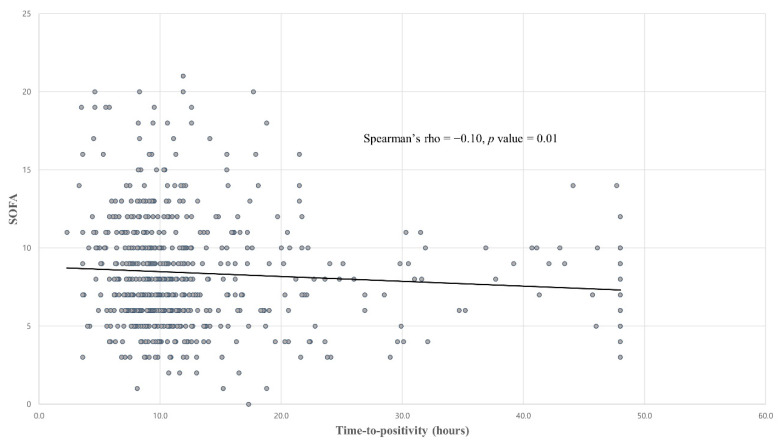
Correlation between the TTP and SOFA scores. Abbreviations: TTP = Time-to-positivity, SOFA = Sequential organ failure assessment.

**Figure 4 antibiotics-10-00683-f004:**
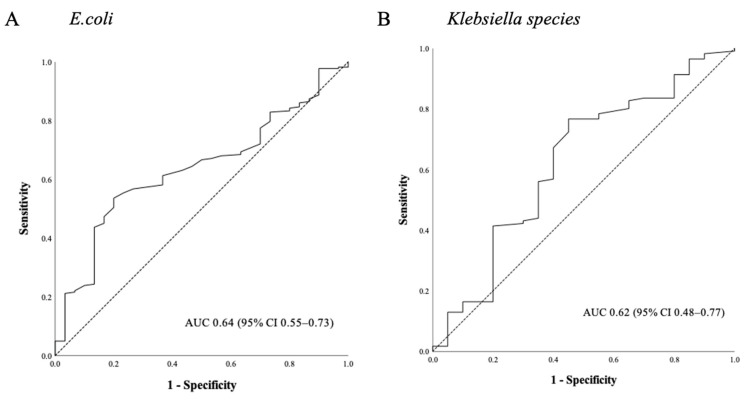
Utility of the TTP ROC curves for predicting the 28-day mortality in septic shock patients with (**A**) *Escherichia coli* or (**B**) *Klebsiella* species bacteremia. Abbreviations: TTP = Time-to-positivity, ROC = Receiver operating characteristic.

**Table 1 antibiotics-10-00683-t001:** Baseline characteristics of septic shock patients according to the blood culture results.

Characteristics	Total (N = 803)	Survivor (N = 674)	Non-Survivor (N = 129)	*p*-Value
Age	67.0 (59.0–75.0)	67.0 (59.0–74.5)	67.0 (61.0–75.0)	0.72
Male	474 (59.0)	395 (58.6)	79 (61.2)	0.58
Past illness				
HTN	265 (33.0)	221 (32.8)	44 (34.1)	0.77
DM	202 (25.2)	169 (25.1)	33 (25.6)	0.90
CAD	79 (9.8)	59 (8.8)	20 (15.5)	0.02
Pulmonary disease	39 (4.9)	33 (4.8)	6 (4.7)	0.91
Malignancy	389 (48.4)	320 (47.5)	69 (53.5)	0.21
Hematologic disorder	50 (6.2)	41 (6.1)	9 (7.0)	0.70
CKD	84 (10.5)	65 (9.6)	19 (14.7)	0.08
LC	12 (1.5)	8 (1.2)	4 (3.1)	0.10
Site of infection				
Unknown	41 (5.1)	34 (5.0)	7 (5.4)	0.86
Lung	109 (13.6)	70 (10.4)	39 (39.2)	<0.01
Urinary tract	143 (17.8)	132 (19.6)	11 (8.5)	<0.01
Intra-abdomen	74 (9.2)	57 (8.5)	17 (13.2)	0.09
Hepato-biliary-pancreas	362 (45.1)	322 (47.8)	40 (31.0)	<0.01
Others	42 (5.3)	35 (5.3)	7 (5.3)	0.44
Initial vital signs				
SBP (mmHg)	93.0 (80.0–112.0)	92.0 (80.0–111.0)	94.5 (75.8–121.5)	0.36
DBP (mmHg)	58.0 (49.0–79.0)	58.0 (49.0–69.0)	59.5 (48.0–74.8)	0.18
PR (/min)	109.0 (90.0–127.0)	108.0 (90.0–126.5)	111.5 (87.5–130.0)	0.48
RR (/min)	20.0 (20.0–22.0)	20.0 (20.0–22.0)	20.0 (20.0–24.0)	<0.01
BT (°C)	37.9 (36.9–39.0)	38.0 (36.9–39.1)	37.4 (36.4–38.5)	<0.01
Laboratory				
WBC (×10^3^/μL)	9.0 (4.5–15.8)	9.1 (5.0–15.8)	8.1 (2.3–15.5)	0.02
Hemoglobin (g/dL)	10.8 (9.0–12.3)	10.8 (9.2–12.3)	10.0 (8.2–12.0)	<0.01
PT (INR)	1.3 (1.1–1.5)	1.3 (1.1–1.5)	1.4 (1.2–1.9)	<0.01
Lactate (mmol/L)	3.3 (2.0–5.6)	3.1 (1.9–5.3)	4.7 (2.7–7.9)	<0.01
BUN (mg/dL)	25.0 (17.0–37.0)	24.0 (16.0–36.0)	31.0 (22.3–44.8)	<0.01
Creatinine (mg/dL)	1.3 (1.0–2.1)	1.3 (0.9–2.0)	1.7 (1.2–2.7)	<0.01
Total bilirubin (mg/dL)	1.5 (0.8–3.3)	1.4 (0.8–3.2)	1.6 (0.8–4.1)	0.40
CRP (mg/dL)	12.3 (5.3–29.4)	11.9 (4.9–20.2)	14.2 (7.6–22.2)	0.06
SOFA score	8.0 (6.0–10.0)	7.0 (5.8–10.0)	11.0 (8.0–14.0)	<0.01
TTP (hours)	10.1 (8.2–13.3)	10.2 (8.3–13.2)	9.4 (8.0–13.6)	0.35
Interventions				
Source control	370 (46.1)	321 (47.6)	49 (38.0)	0.04
Antibiotics escalation	129 (16.1)	105 (15.6)	24 (18.6)	0.39
ICU admission	397 (49.4)	320 (47.5)	77 (59.7)	0.01
Mechanical ventilator	171 (21.3)	100 (14.8)	71 (55.0)	<0.01
RRT	123 (15.3)	69 (10.2)	54 (41.9)	<0.01

Data are presented as a number (%) or as a median with interquartile range. Abbreviations: HTN = hypertension; DM = diabetes mellitus; CAD = coronary artery disease; CKD = chronic kidney disease; LC = liver cirrhosis; SBP = systolic blood pressure; DBP = diastolic blood pressure; PR = pulse rate; RR = respiratory rate; BT = body temperature; WBC = white blood cells; PT = prothrombin time; INR = international normalized ratio; BUN = blood urea nitrogen; CRP = C-reactive protein; SOFA = sequential organ failure assessment; TTP = time-to-positivity; ICU = intensive care unit; RRT = renal replacement therapy.

**Table 2 antibiotics-10-00683-t002:** Frequency of different bacterial strains among the culture-positive results.

Infectious Bacterial Strain	Frequency (%)
*Escherichia coli*	333 (40.8)
*Klebsiella* spp.	191 (23.4)
*Staphylococcus* spp.	60 (7.4)
*Streptococcus* spp.	49 (6.0)
*Enterococcus* spp.	38 (4.7)
*Enterobacter* spp.	36 (4.4)
*Pseudomonas* spp.	23 (2.8)
*Citrobacter* spp.	16 (2.0)
*Acinetobacter* spp.	11 (1.3)
*Clostridium* spp.	10 (1.2)
Etc.	49 (4.9)

Data are presented as a number (%).

**Table 3 antibiotics-10-00683-t003:** Univariate and multivariate analysis for predicting 28-day mortality.

Variables	Univariate			Multivariate		
	HR	95% CI	*p*	Adjusted HR	95% CI	*p*
CAD	1.65	0.84–3.23	0.15	1.79	0.93–3.46	0.08
CKD	1.01	0.50–2.03	0.98			
Lung infection	1.52	0.78–3.03	0.24			
UTI	0.47	0.21–1.05	0.07	0.42	0.20–0.88	0.02
Intra-abdominal infection	1.14	0.51–2.53	0.75			
HBP infection	0.61	0.33–1.11	0.11	0.48	0.29–0.79	<0.01
WBC	1.00	0.98–1.03	0.73			
Hb	0.87	0.79–0.96	<0.01	0.87	0.79–0.96	<0.01
PT(INR)	1.09	0.95–1.24	0.23			
Lactate	1.17	1.09–1.26	<0.01	1.18	1.09–1.26	<0.01
BUN	1.01	0.99–1.03	0.23	1.01	0.99–1.03	0.09
Creatinine	1.00	0.97–1.03	0.96	0.79	0.60–1.03	0.08
CRP	1.01	0.96–1.03	0.48			
SOFA	1.16	1.08–1.24	<0.01	1.17	1.09–1.25	<0.01
Source control	0.73	0.45–1.17	0.19			
Antibiotic escalation	1.02	0.55–1.88	0.96			

Abbreviations: HR = hazard ratio; CI = confidence interval; CAD = coronary artery disease; CKD = chronic kidney disease; UTI = urinary tract infection; PT = prothrombin time; INR = international normalized ratio; CRP = C-reactive protein; SOFA = sequential organ failure assessment; CPSS = culture positive septic shock.

## Data Availability

The dataset used in the study is available from the corresponding author on reasonable request.

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
