# Peer review of "Prognostic Value of the Time-to-Positivity in Blood Cultures from Septic Shock Patients with Bacteremia Receiving Protocol-Driven Resuscitation Bundle Therapy: A Retrospective Cohort Study"

_antibiotics, 2021, doi:10.3390/antibiotics10060683_

Round 1

Reviewer 1 Report

The paper is interesting and well written. It adds new data about the undefined value of TTP for predicting the outcome of the patient. I’d like the Authors to focus on some specific points of clinical interest.

Specific comments

1) the Authors analyzed the value of TTP in specific bacteriemia sustained by Escherichia Coli and Klebsiella, most being part of urinary tract infections. However, from a clinical point of view, as expected good predictors of mortality were lactate and SOFA score (see Table 2), and the prognosis severe when the source of infection was lung (see Table 1). On the other hand, the most common infection was from the hepato-biliary-pancreas tract, followed by the urinary tract. When you stratified the patients by hepato-biliary-pancreas tract, urinary tract and lung infection source (n 139, see Table 1), was the median TTP different between survivor and non-survivor? This information could be interesting as the source of infections is usually determined at hospital admission (Chest X-ray, CT, and so on).

2) Table 1 provides data at admission including creatinine, significantly higher in non-survivor patients. Was increased creatinine the result of acute kidney injury? Do you have data about urine output?

3) In “Interventions” (see Table 1) you do not mention surgical procedures. Were all your patients only medically treated? If not, any difference between medical and surgical patients for TTP as a predictor of 28-day mortality?

Author Response

The paper is interesting and well written. It adds new data about the undefined value of TTP for predicting the outcome of the patient. I’d like the Authors to focus on some specific points of clinical interest.

Specific comments

1) the Authors analyzed the value of TTP in specific bacteriemia sustained by Escherichia Coli and Klebsiella, most being part of urinary tract infections. However, from a clinical point of view, as expected good predictors of mortality were lactate and SOFA score (see Table 2), and the prognosis severe when the source of infection was lung (see Table 1). On the other hand, the most common infection was from the hepato-biliary-pancreas tract, followed by the urinary tract. When you stratified the patients by hepato-biliary-pancreas tract, urinary tract and lung infection source (n 139, see Table 1), was the median TTP different between survivor and non-survivor? This information could be interesting as the source of infections is usually determined at hospital admission (Chest X-ray, CT, and so on).

Response> Thank you for the good suggestions.

The reason why it is difficult to predict the prognosis of patients with septic shock is that the heterogeneity of infection foci and sources is so diverse, and perhaps this diversity is the reason why TTP has limitation in predicting the prognosis for all septic shock patients.

E.coli and Klebsiella bacteremia group were separately selected and analyzed because they were the two most common strains. Unlike the results for the entire group, we found that median TTP was significantly differed between the two groups of survivors and non-survivors.

It is well known that patients’ prognosis varies depending on the source of infection as well as the pathogen. As the review pointed out, septic shock patients with the lung infections showed a worse prognosis than those who suffered from infection with HBP or UTI origin.

When the median TTP was analyzed by stratifying the patients by infection source (lung, HBP, UTI), the following results were as follows.

Table S3. Median TTP according to sites of infection.

Lung infection

Total

(N = 109)

Survivor

(N = 70)

Non-survivor

(N = 39)

P-value

Median TTP (hr)

12.0 (9.2 – 18.8)

12.1 (9.3 – 20.4)

11.8 (9.0 – 15.9)

0.73

Urinary tract infection

Total

(N = 143)

Survivor

(N = 132)

Non-survivor

(N = 11)

P-value

Median TTP (hr)

10.7 (8.6 – 15.1)

10.8 (8.8 – 15.0)

9.4 (7.4 – 21.1)

0.73

Hepato-biliary-pancreas infection

Total

(N = 362)

Survivor

(N = 322)

Non-survivor

(N = 40)

P-value

Median TTP (hr)

9.8 (8.2 – 12.1)

9.9 (8.4 – 12.1)

9.0 (8.0 – 12.6)

0.35

Abbreviations: HTN = hypertension; DM = diabetes mellitus; CAD = coronary artery disease; CKD =

Stratifying patients by the source of infection, the difference in TTP between survivors and non-survivors was not significant

For this, we will add the table as supplement material and explain it in the discussion section.

“When patients stratified by the source of infection, such as lung infection, urinary tract infection and hepato-biliary-pancreas infection, the difference in TTP between survivors and non-survivors was not significant in each group. (12.1 vs 11.8 hours, p = 0.73 for lung infection; 10.8 vs 9.4 hours, p = 0.73 for urinary tract infection; 9.9 vs 9.0 hours, p = 0.35 for hepato-biliary-pancreas infection, Table S3).” (line 252-256)

2) Table 1 provides data at admission including creatinine, significantly higher in non-survivor patients. Was increased creatinine the result of acute kidney injury? Do you have data about urine output?

Response> Thank you for a good point. Numerous previous studies have shown that the occurrence of sepsis-induced AKI suggests poor prognosis.

(Yegenaga I, et al. Clinical characteristics of patients developing ARF due to sepsis/systemic inflammatory response syndrome: results of a prospective study. Am J Kidney Dis. 2004;43(5):817-824.; Bagshaw SM, et al. Septic acute kidney injury in critically ill patients: clinical characteristics and outcomes. Clin J Am Soc Nephrol. 2007;2(3):431-439.)

It can be assumed that the high level of creatinine in the non-survivor group directly/indirectly reflects that the occurrence of AKI, which directly leads to a poor prognosis, as in previous studies.

Unfortunately, the data on baseline creatinine levels or urine output of enrolled patients have not been collected, unlike creatinine levels that are collected routinely when visiting the ER.

Because this study is a retrospective analysis of the registry collected prospectively, there are limitations in defining the difference of creatinine level between survivor and non-survivor based on the AKI criteria suggested by KDIGO.

3) In “Interventions” (see Table 1) you do not mention surgical procedures. Were all your patients only medically treated? If not, any difference between medical and surgical patients for TTP as a predictor of 28-day mortality?

Response> The definition of “Interventions” (n = 370) in this study includes emergent surgery (n = 57), percutaneous drainage (n = 157), endoscopic intervention (n = 124), and removal of infection device (n = 32). After we arbitrarily categorized interventions as surgical, such as emergent surgery and removal of infection device (n = 89), and medical (n = 281), patients treated with surgical intervention showed higher 28-day mortality than medical intervention group (20.2 vs. 11.0 %, p = 0.03). Meanwhile, both groups had similar median TTPs (10.7 [8.5 – 13.2] for surgical vs. 10.0 [8.2 – 13.8] hr for medical; p = 0.45).

Reviewer 2 Report

Authors present an interesting study in which they analyzed TTP relation with septic shock patients mortality. In a very specific group of Escherichia coli and Klebsiella sepsis patients Authors showed that shorter TTP was related with 28-days mortality, as in previous studies (Álvarez R, et al. Time to positivity of blood culture association with clinical presentation, prognosis and ESBL-production in Escherichia coli bacteremia. Eur J Clin Microbiol Infect Dis. 2012. PMID: 22298241; Bo SN, et al. Relationship between time to positivity of blood culture with clinical characteristics and hospital mortality in patients with Escherichia coli bacteremia. Chin Med J (Engl). 2011. PMID: 21362328; Cheng J, et al. Time to positivity of Klebsiella pneumoniae in blood culture as prognostic indicator for pediatric bloodstream infections. Eur J Pediatr. 2020. PMID: 32394266).

However:

1) please use TTP abbreviation in the manuscript; in the abstract TPP can found (line 15);

2) please explain how fungal infections were excluded (line 14 and 76)? quite often antigen tests are used (like galactomannan or B-D-glucan) and tissue samples are not always available (histopathology as the best method of diagnosis), and even when available sometimes negative results can be achieved.

3) please correct BUN abbreviation (line 148) into blood urea nitrogen.

4) The non-survivor group had in general higher risk of worser outcome (Table 1 baseline characteristics), with higher incidence of: hypertension, CAD, malignancy, CKD, what may have huge impact on bacteriemia occurence; can you please discuss that?

5) How you may explain an interesting results: lower creatinine level (HR 0.67, p<0.01) was associated with poorer clinical outcome? (lines 178-181). Or was it GFR? in general we observe much worser outcome of septic patients with high creatinine/low GFR = AKI (Järvisalo MJ, Hellman T, Uusalo P. Mortality and associated risk factors in patients with blood culture positive sepsis and acute kidney injury requiring continuous renal replacement therapy-A retrospective study. PLoS One. 2021 Apr 5;16(4):e0249561. doi: 10.1371/journal.pone.0249561).

Author Response

Authors present an interesting study in which they analyzed TTP relation with septic shock patients mortality. In a very specific group of Escherichia coli and Klebsiella sepsis patients Authors showed that shorter TTP was related with 28-days mortality, as in previous studies (Álvarez R, et al. Time to positivity of blood culture association with clinical presentation, prognosis and ESBL-production in Escherichia coli bacteremia. Eur J Clin Microbiol Infect Dis. 2012. PMID: 22298241; Bo SN, et al. Relationship between time to positivity of blood culture with clinical characteristics and hospital mortality in patients with Escherichia coli bacteremia. Chin Med J (Engl). 2011. PMID: 21362328; Cheng J, et al. Time to positivity of Klebsiella pneumoniae in blood culture as prognostic indicator for pediatric bloodstream infections. Eur J Pediatr. 2020. PMID: 32394266).

However:

1) please use TTP abbreviation in the manuscript; in the abstract TPP can found (line 15);

Response> We corrected typo in the abstract.

2) please explain how fungal infections were excluded (line 14 and 76)? quite often antigen tests are used (like galactomannan or B-D-glucan) and tissue samples are not always available (histopathology as the best method of diagnosis), and even when available sometimes negative results can be achieved.

Response> We fully agree with the reviewer's comment, and it is challenging to completely rule out a fungal infection.

Patients with documented fungal infection (n=66) mentioned in the main article refer to patients whose pathogen have been identified as a fungus in fungal culture (blood, urine or sputum), or patients with beta-glucan or galactomannan positive in blood test.

As prospective study, there was no protocolized guidelines for fungal study though, appropriate microbiological study was conducted in septic shock patient for possible pathogen. Fungal culture and diagnostic test such as beta-D-glucan or galactomannan performed under clinical judgement.

Therefore, among the patients whose culture test results were negative, there may be hidden fungemia patients who did not perform fungus culture by the judgment of the clinician, but this hidden fungemia group is considered irrelevant because they were excluded as bacteremia-negative patients.

There may also be patients with hidden fungal co-infection among bacteremia-positive septic shock patients. We do not believe that this group will have a significant impact on determining the clinical implications of TTP in a group of patients with bacteremia-positive septic shock.

3) please correct BUN abbreviation (line 148) into blood urea nitrogen.

Response> We corrected typo.

4) The non-survivor group had in general higher risk of worser outcome (Table 1 baseline characteristics), with higher incidence of: hypertension, CAD, malignancy, CKD, what may have huge impact on bacteriemia occurence; can you please discuss that?

Response> Thank you for your valuable opinion. We fully agree with the opinion of the review.

As a result of Table 1, there is a difference in underlying diseases between the survival and non-survivor groups, which may well have a significant impact on prognosis.

An additional analysis was conducted to determine whether Bacteremia's incidence differed in underlying diseases, and the following results were found.

Table S4. Baseline characteristics of the septic shock patients according to the blood culture results.

Past illness

Total

(N = 1718)

No bacteremia

(N = 915)

Bacteremia

(N = 803)

P-value

HTN

605 (35.2)

340 (37.2)

265 (33.0)

0.07

DM

425 (24.7)

223 (24.4)

202 (25.2)

0.71

CAD

182 (10.6)

103 (11.3)

79 (9.8)

0.34

Pulmonary disease

108 (6.3)

69 (7.5)

39 (4.9)

0.02

Malignancy

782 (45.5)

393 (43.0)

389 (48.4)

0.03

Hematologic disorder

86 (5.0)

56 (6.1)

30 (3.7)

0.02

CKD

118 (6.9)

68 (7.4)

50 (6.2)

0.32

LC

266 (15.5)

116 (12.7)

150 (18.7)

< 0.01

Data are presented as a number (%)
Abbreviations: HTN = hypertension; DM = diabetes mellitus; CAD = coronary artery disease; CKD = chronic kidney disease; LC = liver cirrhosis.

Based on this information, we will describe the additional results in the discussion part.

5) How you may explain an interesting results: lower creatinine level (HR 0.67, p<0.01) was associated with poorer clinical outcome? (lines 178-181). Or was it GFR? in general we observe much worser outcome of septic patients with high creatinine/low GFR = AKI (Järvisalo MJ, Hellman T, Uusalo P. Mortality and associated risk factors in patients with blood culture positive sepsis and acute kidney injury requiring continuous renal replacement therapy-A retrospective study. PLoS One. 2021 Apr 5;16(4):e0249561. doi: 10.1371/journal.pone.0249561).

Response> Developing acute kidney injury in patients with septic shock is well-known poor prognostic factor. As the reviewer’s recommendations, the association between lower creatinine level and poor clinical outcome was wired in previous result. We make a mistake in logistic regression analysis and corrected typo in table 3.

Round 2

Reviewer 1 Report

I hve no further comments